# Understanding Circadian and Circannual Behavioral Cycles of Captive Giant Pandas (*Ailuropoda melanoleuca*) Can Help to Promote Good Welfare

**DOI:** 10.3390/ani13152401

**Published:** 2023-07-25

**Authors:** Kristine M. Gandia, Elizabeth S. Herrelko, Sharon E. Kessler, Hannah M. Buchanan-Smith

**Affiliations:** 1Psychology, Faculty of Natural Sciences, University of Stirling, Stirling FK9 4LA, UK; 2Smithsonian’s National Zoo, Conservation Biology Institute, Washington, DC 20008, USA

**Keywords:** circadian, circannual, animal welfare, captive, behavior, breeding

## Abstract

**Simple Summary:**

Circadian clocks are evolutionarily adaptive internal clocks that regulate cycles of activity, behavior, and physiological processes. In this study, we used the circadian (24 h) and circannual (across the year) rhythms of behavior of 13 zoo-housed giant pandas to understand their needs and assess welfare. We found that they show changes in the pattern and intensity of behavior cycles based on their life stage and sex. Their circadian activity patterns showed three peaks, like wild giant pandas, including a night-time peak. We found specific cycles of feeding anticipatory activity, sexual-related behavior, and stereotypical/abnormal behavior, which align with the timing of migration in wild giant pandas, and we therefore suggest that this may be how thwarted migration manifests itself in captivity. We also determined the cycles of maternal behaviors, including nursing, and proximity for a mother and cub, providing context to the development of a circadian rhythm in a cub and information on the way a mammalian mother’s circadian rhythm is disrupted during nursing. Overall, our study provides a holistic, evidence-based method that can be applied across captive environments so that staff can better understand the needs of their species and appropriately provide for them, promoting positive welfare and increasing the likelihood of successful breeding and conservation.

**Abstract:**

Circadian and circannual cycles of behavior regulate many aspects of welfare including metabolism, breeding, and behavioral interactions. In this study, we aim to demonstrate how systematically determining circadian and circannual cycles can provide insight into animals’ needs and be part of an evidence-based approach to welfare assessment. We measured and analyzed the observational behavioral data of 13 zoo-housed giant pandas (*Ailuropoda melanoleuca*), across life stages and between sexes, each month for one year using live camera footage from six zoos across the world. Our results indicate that life stage was associated with changes in overall activity, feeding, locomotion, and pacing, and that sex influenced scent anointing and anogenital rubbing. Overall, the circadian rhythms showed three peaks of activity, including a nocturnal peak, as seen in wild giant pandas. We also found associations between sexual-related, stereotypical/abnormal, and feeding behavior, which are possibly linked to the timing of migration of wild pandas, and elucidated the relationship between a mother and cub, finding that they concentrate maternal behaviors to mainly after closing hours. Understanding these cycle patterns can aid animal care staff in predicting changing needs throughout the day, year, and life cycle and preemptively provide for those needs to best avoid welfare concerns.

## 1. Introduction

Circadian rhythms have been recognized across the animal kingdom from birds, reptiles, mammals, amphibians, fishes, and arthropods, including insects [1,2]. The widespread evolution of circadian rhythms suggests that there is an adaptive advantage to possessing one in an environment with cyclical changes, allowing for species to anticipate changes in their environment, like temperature throughout a diel cycle or food availability throughout a circannual cycle, and respond accordingly to maintain homeostasis [3,4]. In the wild, species exhibit cycles of activity throughout a 24 h period and across the seasons. However, in captive environments, research identifying rhythms of activity throughout the day and night and across the seasons is limited to only a few species and is largely unknown within individual zoos, as the care of the animals is limited to the working day and seasonal changes in activity are not closely monitored [5]. It is important to be aware of the various circadian rhythms of captive species, as many aspects of their welfare, including metabolism, breeding, and behaviors, are regulated by their circadian clock [1,6,7,8]. Understanding the diel and annual cycles of behavior and physiology of captive species can help in gaining a holistic view of their needs and thus inform zoo staff on the measures to be taken to promote positive welfare. Our approach to assessing and addressing welfare follows the five domains model [7] by incorporating the assessment of nutrition, environment, health, and behavior on a diel and annual basis to inform an animal’s welfare state.

Animals exhibit life history trade-offs expressed through physiological and behavioral trade-offs to maximize energy and fitness [9]. Energetic imbalances can lead to behavioral changes with implications for welfare [10] or trigger physiological stressors [11,12]. Taking a closer look at cycles of specific behaviors can assist zoo staff in understanding the energetic needs of an animal. Energy maximization occurs when an animal’s fitness is optimized by maximizing the net energy intake for a given time that is spent foraging [13]. Energy maximization goes hand in hand with the circadian system since circadian clocks evolved as a means of synchronizing with the external environment in a way that is most conducive to fitness, which includes efficiently using energy and time [1]. Across several species, one of the clear signs of aging is the inefficient use of energy that results from a disruption of the central and peripheral circadian clocks [14,15]. Therefore, understanding the circadian and circannual cycles of behaviors involved in the conservation and expenditure of energy can assist zoos in deciding how to best provide for the energetic needs of their animals throughout their lifespans, preemptively providing appropriate choices at appropriate times of the day, year, and life stages in preparation for their changing needs.

The optimal foraging theory suggests that animals will choose the type of food, how much time is spent foraging and in which patches, and the speed at which to forage to maximize energy intake and minimize energy expenditure [16]. However, in captivity, zoo staff have nearly complete control of the way an animal will gain energy through food, controlling the kind of food, proportions, and feeding times. This means that zoo staff have a large influence on the regulation of the metabolic rhythm of an animal because feeding regimens are a strong entrainment cue that can supersede the central circadian clock and regulate peripheral tissue clocks and activity [1]. Often, the feeding times and portion sizes given to zoo-housed species are different to those consumed by wild conspecifics [5,17,18]. These altered feeding schedules have the potential to desynchronize metabolic processes with the central clock and alter activity cycles. For instance, a nocturnal mouse placed on a restricted feeding schedule during day hours will display food anticipatory activity (FAA) and be more active around feeding times and less active during normal active hours at night than when fed during the night [19]. Out-of-phase feeding and activity has been associated with shifts in hypothalamic activity that are desynchronized with the central clock [20]. The adverse effects of desynchronizing feeding and metabolic processes with the central clock have been well established [21,22], and are linked to obesity and a variety of disorders involving insulin sensitivity and energetic balance [23].

Aging increases the possibility of desynchrony because the circadian system ages, causing the decoupling of internal cell and/or tissue clocks and, consequently, overt rhythm changes [1,24]. However, this outcome can be ameliorated, and associated diseases can possibly be prevented, by implementing a regular feeding schedule with appropriate kinds of food [14]. This idea is demonstrated in a study by Cincotta et al. [25], where obese aged rats were shown to possess differing circadian rhythms from those of younger, leaner rats. However, when the obese aged rats were injected with metabolic hormones at the times in which the hormones peaked in young lean rats, the obese aged rats began displaying younger phenotypes. The obese aged rats also showed a reversal of age-related insulin resistance and obesity. Being aware of the natural cycles of feeding for captive species and the effects of aging, and designing diets and feeding schedules in accordance with that knowledge, can help zoo staff to avoid the adverse effects of desynchrony and promote optimal health.

Breeding is a behavior that zoos monitor closely to improve conservation efforts; however, successful breeding has been notoriously difficult in some species. In addition, the ability to perform natural sexual behaviors is necessary for animals to feel sexually gratified, contributing to positive affective states [7,26]. Since circadian rhythms regulate and influence many aspects of reproduction, both physiological and behavioral, understanding them can lead to improved breeding efforts and can help in understanding conditions that would allow for these behaviors to be expressed. In humans, the circadian clock has been found to influence every stage of reproduction in females [27]. The circadian rhythm also determines the reproductive cycle for spontaneously and seasonally ovulating animals [28]. For spontaneously ovulating animals, the circadian rhythm optimizes reproduction by ensuring that ovulation occurs at a time of day in which finding a mate and sexual receptivity is likely. For instance, in nocturnal rodents, ovulation occurs at night, when encountering a mate is most likely [28]. For animals with a reproductive season, monitoring the time of year via the circadian clock (i.e., measuring the day length) allows for the animal to ovulate when environmental conditions are most favorable for gestation and rearing [29]. The day length also influences male reproduction. In several species, the male development of testis and better sperm quality are associated with exposure to a long photoperiod [30,31]. However, prolonged exposure to a long photoperiod will then initiate a refractory period, which is ended by the exposure to a short photoperiod [30,31]. Because of the broad influence that circadian rhythms have on reproduction, when creating management and husbandry protocols, being aware of the circadian control of reproduction could potentially increase the rate of successful breeding attempts. This can be achieved by recognizing cycles of breeding, even cycles of behaviors that are not overtly related to mating like increased feeding or general activity, and providing resources to ensure that breeding pairs are synchronized in their behavior and that individual physiological states are optimal. If breeding is successful, zoos must also understand the cycles of dynamics between parents and offspring so that they can adjust their husbandry care to provide the most appropriate resources to the parents so they may effectively raise their offspring.

In captivity, sleep is not often monitored [32]. Though there does not seem to be one consistent benefit to sleep, sleep can be seen as an adaptive state of inactivity [2] with several crucial benefits including homeostatic properties and the improvement of cognitive function [33,34], with prolonged sleep deprivation leading to unavoidable death. Siegel [2] suggests that sleep has evolved to conserve energy and make behavior more efficient in response to the surrounding environment. According to this hypothesis, should sleep be disrupted, either prolonged, shortened, or desynchronized, the efficiency of behaviors and energetic dynamics would be suboptimal and potentially detrimental to the organism. Because of this, monitoring the amount and cycles of sleep and rest can provide insight into the affective states of species and possibly indicate positive affective states or negative ones such as boredom and disease if analyzed in the correct context [35]. For instance, prolonged rest may indicate that an animal is ill and is conserving energy to recover, but can also be suggestive of a positive affective state in a species like the little brown bat, which sleeps for approximately 20 h a day [36]. Sleep health is paramount to general welfare, and therefore, more importance needs to be placed on promoting good sleep, which can only be achieved by understanding what a healthy sleep cycle is for individuals and species.

Migration is a seasonal event that has been widely studied in wild and captive bird species. In migratory birds, migration is triggered by photoperiods of specific length depending on the season [37]. In preparation for migration, bird species will increase fat storages, activate or inactivate reproduction, and molt [38,39]. These changes also occur in captivity despite the inability of the birds to migrate. Migratory inhibition results in a behavior called *Zugunruhe*, from German, which translates to “migratory restlessness”. Captive migratory birds will display increased levels of activity and nocturnal migratory restlessness during periods that coincide with natural migratory behavior [40]. The freedom to perform natural behaviors is part of the gold standard for animal welfare [26]. Therefore, the inability to migrate, despite being physiologically prepared, could indicate negative welfare states. In addition, the increased fat storages accumulated before the migration season may lead to metabolic disorders should they not be used for their intended purpose. This is not only a concern for migratory birds, but across species who migrate such as giant pandas, sea turtles, and certain bat species. However, despite the extensive knowledge of migratory restlessness in captive migratory birds, little is known of the effect of captivity on non-avian migratory species [41]. Investigating these effects by determining the circadian cycles of behaviors associated with migration can lead to suggestions on how to adjust diets during migratory periods or provide opportunities for alternative exercise to promote positive welfare states.

Two of the main goals of reputable zoos are to promote positive welfare of their animals and to conserve species. We aim to investigate behaviors using a holistic approach that could provide insight into the animal’s needs and potential welfare states or provide information on the cycles of behavior that can inform husbandry practices. To demonstrate this holistic approach and to investigate the effects of life stage and sex on captive animal behavior, we investigate the circadian and circannual cycles of captive giant pandas, including a mother and cub pair as a case study. Giant pandas are seasonal breeders that mate between March and April [42]. Giant pandas are also a migratory species, following their food source of bamboo and migrating in the Spring to feed on nutritious shoots [43]. In the wild, giant pandas exhibit three peaks in activity throughout the diel cycle and exhibit a fluctuation in activity levels throughout the year, with a peak in June [44]. Giant pandas being such a specialized species, evolving over time to eat bamboo and adapting their behavior for this food source, makes them a model species for this study as they would presumably need to have well-balanced energy dynamics to sustain themselves. Giant pandas are also ideal for this study because they are a charismatic species and have large popularity worldwide. This means that they have many webcams for observation and can show proof of concept of how monitoring animals on cameras can be very informative while being non-invasive. In addition, pandas are a vulnerable species that are notoriously difficult to breed in captivity. Though large improvements in the captive breeding of pandas have been made, with mate choice being identified as a key factor in successful breeding [45], further understanding of how pandas synchronize their breeding behavior and raise their offspring could greatly improve conservation efforts for the successful breeding of captive pandas.

In this study, we recorded the circadian cycles for one year for male and female captive giant pandas across life stages. We investigated the cycles of general activity and several more specific active behaviors including stereotypical/abnormal behaviors, sexual-related behaviors, and maternal behaviors. These cycles were then compared to each other to infer possible synergistic dynamics between them and to determine possible associations with migration. We hypothesize that life stage and sex have effects on the circadian cycles. Our study cannot determine causality, but providing a holistic view of giant panda behavior can give new insights on the needs of giant pandas and possibly reveal the husbandry and environmental factors that may promote the positive welfare states that zoos want to see in their animals.

## 2. Materials and Methods

### 2.1. Ethics

We received ethical approval for this study from the University of Stirling Animal Welfare Ethics Review Body (Striling, UK, protocol #2084 1591; 30 April 2020). This study also received endorsement from the Association of Zoos and Aquariums, Giant Panda Species Survival Plan (Silver Spring, MD, USA; 5 August 2020). In addition, we submitted research applications to all zoos involved and received approval from the participating zoos’ administrations. We also sent a voluntary questionnaire to keepers asking information on (1) panda identities, (2) cameras, (3) enclosure design, (4) artificial lighting, (5) feeding husbandry, and (6) mating behavior. Four zoos provided some or all of the requested information. The identifying data for the zoos and pandas are anonymized.

### 2.2. Study Animal Selection

The inclusion criterion for a giant panda was whether the panda was in a zoo that had a publicly accessible web camera or a surveillance system that we could be allowed to access. We selected 13 giant pandas (7 females, 6 males, including 1 mother and male cub) from six zoos (Table 1). We identified the pandas by sex and life stage. The life stages follow those determined by Hu [46] of cub (0–1.5 years), sub-adult (1.5–6 years), and adult (6+ years). Hu [46] also included a senior category (20+ years), which we excluded, despite having senior pandas, because of our limited sample size. The panda dam was categorized as being in a maternal life stage, which we considered to be separate from the adult stage. In addition, the two adults (male and female) from Zoo C could not be distinguished during observations; therefore, their data were combined, and their sexes were labeled as unknown for analysis. All cameras were accessible 24 h per day; however, cameras for 6 individuals did not have night vision, so observations for these individuals were limited to daylight hours (Table 1). The remaining 7 individuals had data collected on a 24 h basis.

### 2.3. Behavior Observations

Behavioral observations were completed using the ZooMonitor application [47]. The ethogram was designed to include most of the behavioral repertoire of giant pandas, covering behaviors that indicate positive, neutral, and negative affective states (Table A1). The valences were based on principles that convey the importance of displaying natural behaviors at levels that are conducive to positive welfare [7,48]. Therefore, negative behaviors would be abnormal behaviors, behaviors displayed out of frustration, or high levels of aggression. Positive behaviors are those known to be associated with positive affective states like play, investigation, sexual-related behaviors [49], or natural behaviors that would be performed under natural conditions because they are pleasurable and promote biological functioning [48]. We defined neutral behaviors as those related to maintenance, which are highly dependent on the context and levels at which they are displayed to be considered positive or negative like urinating/defecating, rest/sleep, and locomotion. Principles defining valence of behaviors refer to the species’ typical levels of the behaviors. However, these have generally not been established for most captive species. This study can help to demonstrate a method for determining the species’ typical levels and cycles of behaviors to validate these baselines.

Data were collected through web cameras from December 2020 to November 2021 for all pandas except for the mother and cub. The mother was observed from December 2020 to March 2022, and her cub was observed from March 2021 to March 2022. The cub was aged 6 months when his observations began, providing an opportunity for a case study of how circadian rhythms develop in a cub. Focal sampling was completed for each giant panda using one 10 min sessions with 30 s intervals in each hour to gain an estimate of the behavior in that hour. Each month, data collection began on the 10th day and continued through the end of the month until one daylight or one 24 h cycle was recorded (dependent on the night vision of the camera) for each panda. Observations were performed in real time. Observers were assigned a maximum of 5 consecutive hours of observations, completing a maximum of three 10 min sessions with a minimum of 30 min rest each hour. Observers would determine which pandas needed a session completed in the hour and scan the cameras until they found a panda in view and would complete the 10 min session for the panda in view. Observers were trained to observe all pandas except for the mother and cub, which only the lead investigator observed. In addition, only the lead investigator had access to cameras at Zoo D, and therefore, these pandas were also only observed by the lead investigator. Between all 13 pandas, a total of 2592 ten-minute sessions (432 h) were completed and used for analysis. A graphical representation of the sampling method can be seen in Figure 1.

Vocalizations were not recorded, as some cameras did not have audio. Whether the panda was out-of-sight was also recorded, and a session was only saved if the panda was in sight for 60% of intervals (12/20 intervals) to provide representative data for analysis. Sessions with more than 8/20 out-of-sight intervals were deleted and reperformed.

In total, 13 observers assisted in data collection throughout the data collection period. To produce data that would be used for analysis, observers had to pass reliability testing. Since testing reliability purely from live observations often results in many ethogram behaviors not being evaluated [50], we designed our reliability testing with two stages aiming to cover all ethogram behaviors. The first stage was an ethogram quiz for which the observer had to receive >80%. The second stage was inter-observer reliability using a combination of compilations of short video clips of all behaviors listed in the ethogram, and 10 min recordings from the study pandas mimicking the way the observation sessions would be completed using the web cameras and ZooMonitor. For the short video clips, beeps were placed at variable intervals so that the full repertoire of pandas was covered and indicated when to record a behavior. For the 10 min recordings, observers used ZooMonitor and the 30 s intervals to record behaviors. Observers had three attempts (each attempt had different videos or beeps were changed) to match at least 75% of the recordings from the lead investigator to pass this final stage. The 75% agreement threshold was based on those generally accepted for reliability [51]. Each attempt contained 8–9 video clips with 4–5 beeps each, and 3–4 10 min recordings with 30 s intervals.

### 2.4. Analyses

#### 2.4.1. Variables

Our predictor variables were life stage (cub, sub-adult, adult, maternal), sex (male, female, unknown), hour of day, and season. Definitions of the behaviors we recorded are given in Table A1. We modeled activity, resting/sleeping, feeding, drinking, locomotion, pacing, bipedal standing, anogenital rubbing, and scent anointing in separate models against the predictor variables. Activity was calculated by adding the counts of all behaviors except for resting/sleeping in 10 min sessions. In addition, we refer to sexual-related behaviors as a category, which includes anogenital rubbing, scent anointing, showing interest, and sexual behavior (Table A1).

We also investigated the rhythms of maternal behaviors and proximity for the mother and cub. We determined the rhythms of nursing behavior and non-nursing maternal behaviors (defined in Table A1). The proximity levels were in contact, proximate, and distant. Proximate was categorized as being within 2 body lengths (back end to nose) of the focal panda, and distant was categorized as being greater than 2 body lengths away. Occasionally, the pandas were separated by keepers; therefore, our analysis on proximity excludes this time, since we aimed to investigate the choice in distance between a mother and cub throughout the cub life stage in captivity.

#### 2.4.2. Zero-Inflated Negative Binomial Modeling and Post hoc Pairwise Comparisons

It is common for ecological data to have a high amount of zero values resulting in zero inflation that causes significant biases in analysis because the fit regression becomes flat [52,53]. These zeros are either “true zeros” or “false zeros”. With behavioral data, true zeros are observed from individuals that never display a behavior or when a behavior is not constantly displayed or rare. These zeros are also called structural zeros. False zeros occur from a sampling error, if a behavior is not displayed within the sampling period, or if a behavior is miscoded. These zeros are also known as sample zeros.

Within our data, each of our response variables had a very high percentage of zeros (between 45 and 98%). Our data have both true/structural and false/sample zeros, but mainly zero inflation due to true/structural zeros, which results in overdispersion [53]. Therefore, we needed a model that worked for count rate data (counts of behavior within a 10 min session) and would account for zero inflation due to a combination of factors, and the resulting overdispersion in the data. Negative binomial models are count models that have a parameter that allows for overdispersion [53,54]. The most appropriate model considering the qualities of our data was the zero-inflated negative binomial (ZINB) mixture model. To conduct this analysis, we used the R package *glmmTMB* [55]. In *glmmTMB*, zero-inflated GLMMs have the following three components: a model for the conditional mean (negative binomial in our study), a model for zero inflation, and a dispersion model. The conditional mean and dispersion models analyze positive values using log links. The zero-inflated model describes the probability of observing a true/structural zero that is not generated by the conditional model. The values within the zero-inflated model are constrained between 0 and 1 by applying a logit link [52,55,56]. The overall fit of the ZINB mixed model is determined by all three components. Therefore, when interpreting the results, we must consider the results of all three models. The interpretations of the coefficients for the count model and zero-inflated model are different. A positive coefficient in the count model indicates an increase in the response with an increase in a continuous predictor or in the specified level of a categorical predictor. In contrast, a positive coefficient in the zero-inflated model indicates that a structural zero in the response is more likely to occur with an increase in a continuous predictor or in the specified level of a categorical predictor.

For each behavior in the ZINB mixed model, the zero-inflated model was the same as the conditional model. The categorical variables of season, life stage, and sex were coded within the model using contrast sums. Therefore, each level of the variable was compared against a grand mean within the model, which was the mean of the response variable means at each level of the categorical variable. In addition, Gandia et al. [57] determined that the latitude, temperature, and amount of daylight had an effect on the behavioral cycles of giant pandas. Giant pandas at latitudes higher than the natural range displayed lower levels of activity and a more sporadic pattern of stereotypical/abnormal behaviors. Temperature and daylight had potentially regulatory effects on the patterns of activity. Therefore, although we did not include these variables in our models, our grouping variables were the individual panda nested within zoo to control the location effects. Zoos A, C, and F are located at latitudes that match the natural range of giant pandas, while Zoos B, D, and E are located at latitudes above the natural range towards the poles. Due to our limited sample size, we ran the models using restricted maximum likelihood (REML) rather than maximum likelihood. This is an iterative process, and the final model that is presented is the one with the best estimation. We did not standardize for the time that the pandas were out-of-sight by converting the counts to decimal rates per time in sight because the models require count data, and because, among all sessions, the mean time in sight was 95.6% and the median was 100%.

In addition to the models for all individuals and the selected behaviors, we also conducted ZINB mixed models for female vs. maternal female activity. This model only had hour, season, and sex as predictor variables, with sex having the activity levels of a female or maternal female panda. The season was coded using contrast sums, while sex was coded using the standard dummy coding, where the female level served as a control. For the mother and cub pair, we also modeled nursing, maternal behaviors (all maternal behaviors except for nursing, as described in Table A1), and proximity in separate ZINB mixed models against the predictors of hour and season. The models for nursing and proximity were mainly used to determine whether these behaviors and proximities associated with a nursing mother would change overtime as the cub became closer to being weaned. Therefore, since the mother was observed for over a year, the season in these models was coded by year (Winter 20–21, Spring 21, Summer 21, Autumn 21, Winter 21–22, Spring 22), with Winter 20–21 only having the mother’s data, and Spring 22 only comprising the month of March. The season was coded using contrast sums. The grouping variables of the panda nested within the zoo remained the same along with the use of REML.

The R program *glmmTMB* allows for two kinds of negative binomial models, including one that models the count data using linear regression, and another that uses quadratic regression. The type of the regression chosen was dependent on which one produced a better fit, determined using the AIC score and dispersion parameter. The kind of regression used for each variable is listed in Table 2.

A post hoc analysis of pairwise comparisons of estimated marginal means (least-squares means) were conducted on the season, sex, and life stage variables using the R package *emmeans* [58] to determine any significant differences between variable levels. Multiple comparisons were controlled using the Tukey method. Test-wide alpha was set at 0.05.

#### 2.4.3. Continuous Wavelet Transform and Wavelet Coherence Analysis

To extrapolate more information on mother and cub behavioral cycles throughout the year, the synchronization between mother and cub, and how animals synchronize sexual-related behaviors (all four sexual-related behaviors listed in Table A1), we used continuous wavelet transform and wavelet coherence analyses. Wavelet transform is a time series analysis where a signal is transformed into a wave with a zero mean that is expanded and localized in both frequency and time. This allows for the detection of periodic patterns of a time series in both time and frequency domains while controlling the random background noise in the signal. Continuous wavelet transform is useful for analyzing localized intermittent oscillations in a single time series, allowing for the identification of cycles within cycles (e.g., 8 h cycle within a 24 h cycle). This analysis is ideal for asking questions about how a cub’s undeveloped circadian rhythm changes over time, and the extent to which a mother’s circadian rhythm is disrupted and recovered while having a dependent cub. Using the continuous wavelet analysis allows us to identify whether the mother and cub display similar cycle lengths overall and whether the amplitude of these cycles is similar between the two.

Using continuous wavelet transform, we can also compare two wavelets and conduct a wavelet coherence analysis to determine how two time series are related to each other. Using wavelet coherence analysis, we can examine whether regions in time frequency space with a similar high power have a sustained phase relationship, possibly suggesting a relationship between the signals [59]. Essentially, we can determine areas of correlation between the two wavelets. This type of analysis can be used to determine when and to what extent the mother and cub were synchronized in their behavioral cycles throughout the period of observation where both the mother and cub were recorded (excluding first few months where only mother’s behavior was recorded). Wavelet coherence can also be used to determine whether breeding pairs in zoos are synchronizing their sexual-related behaviors. For our analysis, we conducted a wavelet coherence analysis between the mother and cub pair and for sexual-related behaviors between breeding pairs with 24 h data (2 pairs, Zoos B and D).

To conduct the analysis, we used the MATLAB Wavelet Toolbox developed by Grinsted et al. [59]. Our data are non-stationary, so we used a continuous wavelet analysis with the Morlet wavelet and a scale resolution of 10 scales per octave, as suggested by Grinsted et al. [59], since these settings provide a good balance between time and frequency localization. Our sampling period was set as Δt = 1 h. We also decided to conduct continuous wavelet transform and wavelet coherence analysis, because these analyses were used by Zhang et al. [60] to address similar questions regarding wild giant pandas with activity data recorded using radio collars. We followed their analyses as closely as possible so that our data could be compared to the results found in wild pandas, including one mother panda. However, because of our sampling method, our behavior signals are composed of consecutive representative 24 h periods in each month. Therefore, we could only determine the patterns of circadian cycles and the annual cycle of those circadian cycles, but could not infer anything about rhythms with a month-long period.

## 3. Results

The model summaries for each behavior category can be seen in Table 2. Each of the behavior category’s ZINB mixed models had significant coefficients in either the count model or the zero-inflated model, or both. The full results of the significant coefficients from the general behavior ZINB mixed models can be seen in Table 3. The full results of the significant coefficients from the female activity and maternal case study models can be seen in Table 4. Within these tables, the coefficients with trends towards significance (α < 0.1) are displayed. However, we only discuss the results that reached the α = 0.05 level of significance. We describe the results for both components, addressing the count model, the zero-inflated model, and then the post hoc pairwise comparisons. The hour of the day was significant for all general behavior ZINB mixed models within the count, zero inflation, or both components (Table 3). For the models including only the mother and cub, we only discuss the pairwise comparisons for the seasons since we are mainly interested in the difference between the seasons and hours, but pairwise comparisons of hours are not possible; however, the full list of significant coefficients for the season and hour can be found in Table 4.

### 3.1. Activity

The statistical results are shown in Table 3 and Table 4, and the activity is graphically represented in Figure 2a. The count model for activity shows a slight decrease in activity in the Summer and a slight increase in the Winter when compared to the grand mean of all seasons. There were also lower levels of activity in the cub and higher levels of activity in the mother when compared to the grand mean of all life stages. The zero-inflated model indicates that in the Autumn and Summer, activity was more likely to be zero, while in the Spring, the pandas were more likely to show some level of activity. In addition, the mother’s activity was less likely to be a zero. This result, paired with the mother’s result in the count model, suggests that she was more consistently active and at higher levels than other individuals, which can also be seen in Figure 2a. The pairwise comparisons also reveal that the mother had significantly higher activity levels than the sub-adults (0.165, z.ratio = 2.799, *p* = 0.026). The activity among all pandas was also lower in the Summer compared to the Winter (−0.120, z.ratio = −3.045, *p* = 0.012). For resting/sleeping, the count model reveals a slight decrease in rest/sleep in the Spring compared to the grand mean for all seasons. Within the zero-inflated model, all the coefficients are significant except for either sex (Table 3). The post hoc analysis did not reveal any significant pairs.

Feeding accounts for 70.6% of active behavior overall and, consequently, has similar circadian and circannual rhythmicity to overall activity (Figure 2b). The count model found higher levels of feeding in the adults and lower levels in the cub compared to the grand mean of all life stages. The zero-inflated model indicates that feeding was more likely to be at zero in the Autumn and Summer, and more likely to be a positive integer in the Winter and Spring. In addition, the cub was more likely to display true zeros for feeding (does not include nursing behavior). This result, combined with the result from the count model for the cub, indicates that the cub fed less often and likely at a lower intensity than the other individuals. The pairwise comparisons confirm this, with the cub showing significantly less feeding behavior than his mother (−0.525, z.ratio = −2.679, *p* = 0.037), the sub-adults (−0.426, z.ratio = −2.709, *p* = 0.034), and the adults (−0.553, z.ratio = −3.937, *p* < 0.001), and this can also be seen in his circadian and circannual cycles of feeding in Figure 2b.

For drinking, the count model and pairwise comparisons produced no significant coefficients or differences. However, the zero-inflated model indicates that it was more likely for drinking to be at the zero level during the Summer and less likely during the Spring. The circadian cycle of drinking seems to show the largest peak among all individuals in the afternoon (Figure 2c).

For locomotion, the count model indicates an increase in locomotion in the Winter and a decrease in the Summer compared to the grand mean of all seasons. In addition to the increased levels of locomotion in the Winter, the zero-inflated model also indicates that locomotion was more likely to be at zero levels in the Winter. These two results together may suggest that pandas have more intense but less frequent bouts of locomotion in the Winter. These results for the seasonal variation in locomotion are further supported by the pairwise comparison, showing significantly more locomotion in the Winter compared to the Summer (0.421, z.ratio = 2.892, *p* = 0.02), and can be seen in Figure 3a. The zero-inflated model also suggests that locomotion is more likely to be a positive integer in the Spring. In addition, the sub-adults are more likely to show true zeros for locomotion, while the cub is more likely to show some level of locomotion.

### 3.2. Stereotypical/Abnormal Behaviors

We chose to analyze pacing and bipedal standing because these were the two most common stereotypical/abnormal behaviors, with pacing accounting for 89.3% of all stereotypical/abnormal behaviors, and bipedal standing accounting for 4.9%.

Within the count model, we saw that the mother displayed more pacing compared to the grand mean across life stages. In addition, pacing increased in the Spring compared to the grand mean of all seasons. The zero-inflated component also indicated that during the Spring, the pandas were less likely to show true zeros for pacing, or rather, more likely to show a positive integer, which may partially account for the increased levels indicated in the count model as they were more consistently showing pacing. Conversely, in the Summer, the pandas were more likely to show true zeros for pacing. These results on the seasonality of pacing were further supported by the post hoc pairwise comparisons showing that there was significantly more pacing in the Spring compared to the Summer (z.ratio = 2.424, *p* = 0.0725). Also, despite being locomotor pacing, the circadian and circannual rhythms of pacing did not match those of locomotion (Figure 3).

The bipedal standing count model shows that there was significantly more bipedal standing in the Spring compared to the grand mean of all seasons. The zero-inflated model indicates that a true zero in bipedal standing is more likely in the Autumn and less likely in the Winter. In addition, the pairwise comparisons indicate that bipedal standing in the Spring is displayed at significantly higher levels than in the Summer (1.699, z.ratio = 2.965, *p* = 0.016) and Winter (1.614, z.ratio = 3.287, *p*= 0.006). Although no results for life stage were significant for bipedal standing, we displayed the life stages for comparison with pacing, as this behavior was often performed in conjunction with pacing. Bipedal standing also displays a similar early morning peak to pacing and increases in the Spring (Figure 3c).

### 3.3. Sexual-Related Behaviors

During the study period, there was no mating witnessed. Therefore, our analysis consists of investigating the behaviors of scent anointing and anogenital rubbing because they are thought to be ways of signaling home range occupation, competitive ability, and fitness [61,62,63].

Within the count model for scent anointing, the males displayed a decreased level of scent anointing compared to the grand mean observed across both males and females. Only the hour was significant within the zero-inflated model. The post hoc analysis revealed no significant differences between the category levels. The circadian rhythm of scent anointing reveals that the females scent anoint at night when the males are not scent anointing (Figure 4a); however, it should be noted that those females are sub-adults.

Anogenital rubbing was modeled with only sex, season, and hour, because adding age overparameterized the model. The count model suggests that anogenital rubbing significantly increases in the Spring and decreases in the Summer compared to the grand mean. The zero-inflated model also indicates that during the Summer, anogenital rubbing was less likely to be at the zero level, or rather, the pandas were more likely to show some level of anogenital rubbing. This result, combined with the decrease in the Summer seen in the count model, may suggest that anogenital rubbing may occur more consistently in the Summer, but at low levels. The zero-inflated model also indicates that during Autumn, the pandas were more likely to have true zeros for anogenital rubbing. In addition, the females were more likely to have true zeros for anogenital rubbing, while the males were less likely to have true zeros. The post hoc pairwise comparisons show that there was significantly less anogenital rubbing in the Summer compared to Autumn (z = −3.126, *p* = 0.01), Spring (z = −4.352, *p* < 0.001), and Winter (z = −3.602, *p* = 0.002). This decreased level of anogenital rubbing in the Summer can be seen in Figure 4b. The circadian rhythm of anogenital rubbing seems to indicate that most anogenital rubbing occurs during daylight hours.

To investigate to what extent a breeding pair synchronizes their sexual-related behaviors (Table A1), we conducted a wavelet coherence analysis on two breeding pairs that had 24 h data (Figure 5). Figure 5a,b presents the scale-averaged wavelet spectrum for each individual, a smoothed signal averaged across all cycle lengths that, given our sampling method, essentially produces a circannual rhythm of circadian rhythms. Both females, despite being in different zoos, displayed similar patterns in their sexual-related behaviors, displaying three progressively more intense peaks through the year (small, medium, then large peak). Both females displayed a small peak in January and a medium-sized peak in the mating season (March–May). Meanwhile, the males displayed more small/medium peaks throughout the year and one large peak. Both males displayed their largest peaks towards the end of Autumn or in the Winter. The male from Zoo B showed consistent but small peaks throughout the breeding season. These similarities between the females and males of the two zoos is interesting; however, more breeding pairs would have to be examined to determine if these kinds of patterns are consistent across females and males.

The two breeding pairs had different coherence patterns in their sexual-related behaviors. The coherence of cycle lengths greater than 24 h cannot be interpreted because we only gained an estimate of one day in each month, so cycles greater than 24 h in our data do not have real-world equivalents. The pair from Zoo D were in phase in their 4 h cycle lengths in January, but were out of phase during the mating season in their 8 h cycle lengths (Figure 5c). However, they had a strong coherence across all cycle lengths in November, which can also be seen with their matching large peaks in their scale-averaged spectrums. The breeding pair from Zoo B did not have in-phase sexual-related behaviors at any point in the year, but during the breeding season around April, there was coherence in their 24 h cycle lengths, with the female signal leading and the male signal lagging, which then switches between August and September, with the male signal leading and the female signal lagging. There was also some coherence between November and December for cycle lengths between 8 and 16 h.

### 3.4. Mother and Cub Behaviors

When modeling the activity of the maternal female against all other females, there was no significant difference between the maternal female and the other females. However, given that this model compares one individual to six, it is important to note that the difference shows a trend towards significance (0.105, z = 1.75, *p* = 0.08). The circadian and circannual rhythms also show the maternal female displaying consistently higher average activity levels than the other females, with some areas of no overlap between 95% confidence intervals (Figure 6).

When modeling the non-nursing maternal behaviors, the models would not run, which is likely due to insufficient data. However, we still found it important to visualize the circadian and monthly rhythms of these behaviors (Figure 7a). For nursing behavior, there were no significant differences between the seasons, and only the hour was significant within the zero-inflated model. However, it is clear that within the circadian rhythms of both the general maternal and nursing behaviors that these behaviors were mostly displayed after zoo opening hours (Figure 7a,b, left panel), with only one peak seen in the afternoon for nursing. These results are complemented by the circadian rhythm of proximity, where the mother and cub spent the most time in contact or proximate after zoo hours and spent a larger proportion of time being distant during hours when the zoo was open (Figure 8).

The circannual rhythm of nursing did not show any changes over time, but it should be noted that the cub was not fully weaned (i.e., separated from his mother and eating independently) until 11 months after the observations ended. The pairwise comparisons for the proximity of the mother and cub indicate that they were significantly more in contact in Summer 21 compared to Autumn 21 (0.756, z.ratio = 2.857, *p* = 0.049) and Spring 22 (1.016, z.ratio = 3.141, *p* = 0.021) (see Figure 8, right panel). The pair was also proximate significantly less in Autumn 21 compared to Winter 20–21 (−0.722, z.ratio = −3.034, *p* = 0.029), Spring 21 (−0.742, z.ratio = −3.608, *p* = 0.004), Winter 21–22 (−1.155, z.ratio = −6.480, *p* < 0.001), and Spring 22 (−1.066, z.ratio = −4.721, *p* < 0.001). They were also proximate significantly less in Summer 21 compared to Winter 20–21 (−0.699, z.ratio = −2.865, *p* = 0.048), Spring 21 (−0.718, z.ratio = −3.452, *p* = 0.007), Winter 21–22 (−1.131, z.ratio = −6.23, *p* < 0.001), and Spring 22 (−1.042, z.ratio = −4.496, *p* < 0.001). The model for distant proximity displayed a model convergence warning, which indicated that the best fitting model was likely not found, and therefore, the AIC and BIC scores were not produced. However, the pairwise comparisons indicate that the mother and cub spent significantly less time being distant in Spring 22 compared to Spring 21 (−0.254, z.ratio = −3.281, *p* = 0.013), Summer 21 (−0.222, z.ratio = −2.867, *p* = 0.048), and Autumn 21 (−0.229, −2.952, *p* = 0.037). They also spent significantly more time being distant in Spring 21 compared to Winter 21–22 (0.146, z.ratio = 2.763, *p* = 0.063). These results do not support our prediction that the mother and cub would gradually spend more time being distant as more time passed.

We compared the activity cycles of the mother (16 months of data) and cub (13 months of data) using continuous wavelet transform. We found that they displayed similar clear cycle lengths of activity as follows: ~3 h, 5 h, 8 h, 12 h, and 24 h. The power of the cycles (a measure of the consistency of magnitude through time) was similar between the mother and cub for the 3 h, 5 h, and 8 h cycles. However, the cub had a stronger 12 h cycle, and the mother had a stronger 24 h cycle, with the mother showing an increasing magnitude of her 24 h cycle towards the end of the observation period (red square in Figure 9a). In addition, we also conducted a wavelet coherence analysis to determine when the activity for the mother and cub was synchronized (Figure 10). We found that the mother and cub periodically had their 4–8 h cycles synchronized throughout the observation period. Interestingly, their 24 h cycles showed strong synchronization between May and July 2021, which was not when the mother had a consistent and strong 24 h cycle.

## 4. Discussion

Our results show the holistic perspective we gain of behavior when observing the circadian and circannual cycles. Our hypotheses that life stage and sex would influence these cycles was supported, as we found significant coefficients and/or differences between the life stages for overall activity, resting/sleeping, feeding, locomotion, and pacing, and found significant coefficients regarding sex for scent anointing and anogenital rubbing. There were no differences among life stages or sex for drinking nor bipedal standing. However, all behaviors displayed daily and seasonal patterns. Understanding these cycle patterns can aid animal care staff in predicting the animals’ changing needs throughout the day, year, and life cycle, and preemptively provide for those needs to best avoid welfare concerns. Though the sample size in this study is relatively small, we believe that the conclusions are still meaningful and can be built upon in future studies.

### 4.1. Holistic View of Active and Inactive Behavior Provides Insight into Energy and Behavioral Dynamics

Investigating the cycle of overall activity is a simple way of gaining information and context on an animal’s routine, circadian clock health, and sleep health. In relation to routine, it was clear that our study pandas all followed a similar activity cycle, with the following three peaks in activity: the largest peak in the early morning, a second peak spread throughout the afternoon, and a third peak in the middle of the night. An earlier study investigating the cycles of activity in captive pandas at the Wolong National Nature Reserve, Sichuan, CN found two peaks of activity in the daytime, which is similar to our results for the daytime peaks [64]. However, in a study completed on five wild pandas fitted with radio collars collecting 24 h data, it was discovered that giant pandas have three peaks including some nocturnal activity [44], therefore demonstrating that captive pandas follow similar rhythms to their wild counterparts. The inclusion of nocturnal activity provides a complete picture of these cycles and allows us to infer with more surety what a homeostatic cycle looks like in pandas.

In addition to this similar result in the circadian rhythm, we also found very similar results in the circannual rhythm when compared to wild pandas. Zhang et al. [44] found that activity peaked in the Spring, was reduced in the Summer and Autumn, then increased again in the Winter. Our models and figures illustrated very similar cycles, where the highest activity across individuals was seen in the Spring. These similar results between giant pandas in the wild and those in captivity depict the strength of circadian clocks and indicate the importance of considering them when understanding the behaviors, and consequently, the needs of a species. Since circadian clocks are an adaptive mechanism for synchronizing the internal environment with the external environment to optimize functionality and energy, they inherently play a large role in animals’ welfare.

Resting/sleeping behavior is important to investigate when determining the welfare state of an animal because it can be one of the clearest indicators of circadian system health. Our results indicate that pandas rest/sleep slightly less in the Spring, which is perhaps in correlation to the migratory or mating period, though further investigation would be needed. They also generally seem to rest/sleep at midday, and then again at night, mixed with nocturnal activity. The adults and maternal female tended to be more likely to display zero rest/sleep, while the cub and sub-adults were more likely to display positive integers. This could indicate that adults have more sporadic rest/sleep when compared to sub-adults, which can be supported by the larger variability in sleep that is seen at night in adults. Though definitive conclusions cannot be drawn, the sleep/wake cycle is known to be disrupted by aging as a result of general aging of the circadian system, resulting in desynchrony with the external environment and the decoupling of internal circadian clocks, often resulting in irregular sleep or arousal thresholds [15,24,65]. In addition, sleep is crucial to the functioning of innate immunity, as many cycles of immune responses and DNA repair are dependent on sleep [66,67]. Therefore, tracking sleep in captive environments is important to assess the overall health of an animal, especially older animals, and could assist in identifying early symptoms of underlying disorders that result in circadian dysregulation.

All zoo animals have a need for a suitable diet, which is usually entirely provided by the zoo staff. In order to provide appropriate food for a species, we have to understand their metabolic and energetic needs and how these cyclically change throughout the day, the seasons, and their lifespan. Investigating the rhythms of more specific major active behaviors, including feeding and inactive behaviors, can help in obtaining this understanding. Giant pandas are peculiar in that they have a very low energy expenditure relative to other species [68]. This is likely an evolutionary trait because of their diet of bamboo, which their gut is not well adapted to obtain high levels of nutrition from, since they are incapable of ruminating to draw more nutrients. Therefore, pandas must eat large amounts of bamboo to compensate, spending higher proportions of time foraging and feeding. We found that pandas spend 70.6% of their active time feeding, with adults showing significantly higher proportions of their time feeding compared to the grand mean, and the cub showing lower proportions. Nie et al. [68] found a significant effect of body mass on the energy expenditure of pandas, with a larger body mass coinciding with a higher daily energy expenditure. With presumably larger body masses, adults would need to feed more often to maintain their weight, given their higher energy expenditure. The cub spending significantly less time feeding would be due to his smaller body mass and the fact that he was nursing throughout the observation period, which was not coded as feeding. These kinds of data can be used to determine the amount of food needed for different life stages and could also be used as indicators of a transition into new life stages. The cycles of feeding are later explained in the context of migration and in relation to stereotypical/abnormal behaviors.

Locomotion is an activity that depletes energy more than other activities. Therefore, according to the optimal foraging theory, this behavior would have to occur in an efficient manner given the low-quality bamboo food source of pandas. Our results suggest that locomotion is an activity limited to the day, with very low levels occurring at night. Mainly, locomotion seems to reach its highest level at midday and then steadily decreases into the late afternoon. This may be to conserve energy and to avoid displaying an energetically taxing behavior in the hotter times of the day. We also found that locomotion significantly increases in colder temperatures in the Winter. Since a panda’s pelage is able to efficiently retain body heat, it would be an ideal time to increase locomotion. In addition to these results, it is known that wild pandas show a very low mean movement speed [68], further maximizing their energy.

The circadian and circannual cycles of drinking have not previously been determined in giant pandas. The circadian rhythm of drinking seems to have several peaks throughout the day and night that match the peaks seen in activity, with the largest peak of drinking among all individuals occurring in the later afternoon. This peak could be in response to completing most of the energy expenditure of the day that occurs mostly in the first half of the day. We also found that drinking was more likely to be at the zero level during the Summer and more likely to be a positive integer during the Spring. These shifts in drinking coincide with the seasonal decreases and increases in activity in the Summer and Spring, respectively. Like many other organs, the renal system has a circadian clock that regulates water retention and the homeostasis of electrolytes [69]. Having a baseline for the comparison of individual- or species-level drinking rhythms could help in the early identification of issues related to water retention and homeostasis if a change in the amount or timing of drinking occurs. These rhythms could be paired with the rhythms of urination and defecation. Our sampling combined urination and defecation, making it difficult to conduct meaningful comparisons with drinking.

We investigated the rhythms of scent anointing and anogenital rubbing as sexual-related behaviors because they are associated with signaling home range occupation, competitive ability, and fitness [61,62,63]. The circadian rhythm of scent anointing displayed males concentrating their scent anointing to daylight hours, while the females showed two large peaks of scent anointing in the night in addition to lower frequencies in the day. However, it should be noted that these large peaks were displayed by sub-adult females, and it is unknown whether they would continue displaying this behavior once they reached sexual maturity. Our results were contrary to a previous finding that male giant pandas scent anoint more often than females [62]. It is important to highlight, however, that two of the six males in our sample likely had reduced sexual competitiveness because one adult male in our sample was castrated for medical reasons, and another is a cub. The male pandas in the study conducted by Charlton et al. [62] were seen to anoint preferentially with strong odors, likely as an olfactory signal of competitiveness to other males [62].

The seasonal cycle of scent anointing that we found may support the competitive signaling hypothesis, since we recorded that the males scent anointed mainly in January and March, leading into the mating season, while the females show a similar peak in January, but have their largest peaks in May and August. Though again, it is important to note that the peak in August seen in the females was entirely seen in sub-adult females, so this may be an indication of sexual immaturity [62,70].

Anogenital rubbing and scent anointing have similar functions and are presumed to be related to each other. They are often displayed in response to each other in carnivores, including other bears and giant pandas [62,71]. Indeed, we did see a similarity in the cycles of anogenital rubbing and scent anointing and their frequencies. Anogenital rubbing occurred at significantly higher levels in the Spring compared to the grand mean, and can be seen to increase in both males and females, supporting the hypothesis that it is related to sexual signaling since its highest levels are shown in the breeding season [72]. Both signals also showed decreases in the Summer, though the levels were also less likely to be zero in the Summer like in scent anointing, suggesting that both behaviors are shown more consistently but at lower intensities in the Summer. We did not find significantly higher or lower averages of anogenital rubbing by sex; however, the females were more likely to display true zeros for anogenital rubbing, while the males were more likely to display some level of anogenital rubbing. These are similar to the results obtained in other studies on giant pandas that found that males scent mark more often than females [62,63], and could be an indication of male–male competition.

### 4.2. Relationships between Migratory, Sexual-Related, Feeding, and Stereotypical/Abnormal Behavioral Cycles

Migration is an inherent behavior in many species [37,40]. The phenomenon of migratory restlessness has mainly been studied in birds. It occurs in species that are both obligate migratory birds [73] and facultative migratory birds [8]. However, migratory restlessness in non-avian captive species has not been given much attention. Though migratory restlessness has not been investigated, similar questions have been asked of captive carnivores with large home ranges. It was found that stereotypical pacing frequency was positively correlated with home range size and average chase distance of captive carnivores [74,75]. It could be that in non-avian migratory species, migratory restlessness is expressed through stereotypical pacing or other stereotypical/abnormal behaviors.

Giant pandas are a migratory species; their migration patterns have been investigated in the wild, and evidence suggests that their main motivation for migration is to follow the emergence of nutritious bamboo shoots, though there are also effects of solar radiation and habitat preference [42,44,60,76,77]. Wild pandas initiate migration from mid-April to early June, migrating within several days to their Summer range and returning over several weeks from early September to October [42]. The fast migration at the start of the Spring coincides with the emergence of bamboo shoots across the elevational gradient, with shoots at higher elevations showing a gradient delay in emergence [77]. Importantly, the initial period of migration also coincides with the breeding season in the Spring. The same study used GPS collars and was also able to note that despite the elevational migration pattern across the pandas, the individual paths were distinct, and they were associated with the possibility that pandas were also seeking mates.

In captivity, pandas will presumably have two concurrent motivations to migrate, which are to find emerging bamboo shoots and to breed. Our investigation into the circadian and circannual rhythms of feeding, sexual-related behaviors, and stereotypical/abnormal behaviors can help us begin to elucidate the connections between these behaviors and migration in captive giant pandas. The most common stereotypical/abnormal behaviors recorded in our study are locomotor pacing and bipedal standing. The circadian and circannual rhythms of these two behaviors were very similar. This is not surprising, as bipedal standing is often incorporated into pacing behavior. The circadian rhythms of both behaviors showed clear peaks in the early morning hours and decreased levels throughout the rest of the day, with hardly any display of either behavior through the night. For the circannual rhythms of pacing and bipedal standing, the pairwise comparisons for both behaviors indicate that there were increased levels in the Spring, which is the time of initial migration and the breeding season, that were significantly higher than in the Summer, respectfully.

The circannual rhythms of pacing and bipedal standing closely resemble that of anogenital rubbing, with the peaks for all three behaviors being in the Spring. Anogenital rubbing and handstand urination was also commonly incorporated into pacing. With anogenital rubbing being the behavior that is more closely linked to sexual signaling, it is likely not coincidental that these behaviors were displayed together, and could be an indication of their intent when incorporated into pacing. Previous studies have also found a relationship between stereotypical behavior and sexual-related behavior in pandas, with one study finding that captive males show more locomotive stereotypes than females, and that this behavior is correlated with reproductive performance [78]. Gandia et al. [57] also conducted a wavelet coherence analysis of stereotypical/abnormal and sexual-related behavior signals and found coherence between the cycles. Interestingly, wild male pandas show increased locomotion in the mating season when compared to females [42], but our study of locomotion displays very different results regarding the circannual rhythm of locomotion, with the highest levels being in the Winter. This disparity between wild and captive locomotor activity could suggest that captive pandas replace locomotion with locomotor pacing to fulfill the unmet need of migration and longer travel periods during the mating season.

Although the circannual rhythms of anogenital rubbing and stereotypical/abnormal behavior were similar, the circadian rhythms were quite dissimilar, which leaves a gap in the explanation for the drivers behind the daily rhythmicity of stereotypical/abnormal behavior. One explanation for the clear rhythmicity of an early morning peak in both pacing and bipedal standing (an anticipatory behavior displayed at keeper doors where pandas will stand on hind legs to peek through windows into keeper areas) could be feeding anticipatory activity (FAA). Pandas across all the studied zoos were left with bamboo overnight and would receive fresh bamboo in the mornings when the keepers arrived. Predictable feeding schedules cause animals to display anticipatory behaviors, which may signal stress should their frequency increase in response to the predictable schedule becoming delayed [79,80,81]. Since stereotypical/abnormal behaviors may be a sign of anticipation [81], the early morning peak in the feeding behavior and both pacing and bipedal standing, which do not coincide with the sexual behavior in the pandas, may indicate anticipation in relation to the keeper and fresh food arrival.

This result is similar to that found in the study investigating rhythms of activity in captive pandas at the Wolong Reserve, where the pandas were seen to have two peaks of daytime activity that coincided perfectly with the feeding times [64]. Additionally, in a study on captive brown bears, the bears were placed on a reversed feeding regimen, being fed during dark hours, resulting in their feeding schedule completely switching to being nocturnal, evidencing FAA [82]. Displaying FAA may be a signal of circadian desynchrony in zoo animals if the feeding times do not coincide with the natural circadian rhythm of feeding, and FAA should be monitored in relation to welfare. It is especially important to consider since not just feeding time, but also diet composition, can override and desynchronize the central circadian clock, with chronic effects leading to premature aging [14,83,84,85]. Pandas display a seasonal preference for plant parts in both the wild and in captivity, with a strong, natural inclination towards shoots in the Spring for wild individuals. One possible way of easing FAA could be to provide pandas with appropriate amounts of their seasonal preferences, taking into account the wild and innate rhythms of preference for plant parts, and providing enough of these parts to sustain pandas while the keepers are not present.

Our results may indicate a combined effect of anticipation for feeding and breeding opportunities. Zoo staff can use the cycles and paired behaviors to infer the needs that the panda wants to express. Pacing paired with bipedal standing can be an indication of food anticipatory activity, while pacing combined with anogenital rubbing can be a desire for breeding opportunities. Observing full cycles and comparing them all together allows for new associations between behaviors to be made. Though the associations between stereotypical/abnormal behaviors, feeding, and mating cannot be generalized to all species, the method in which these associations are investigated can be generalized. Circadian and circannual rhythms provide a fuller picture of behavior, and investigating them provides new insights into the possible motivations behind behaviors.

### 4.3. Case Study of Mother and Cub

Previous studies on giant pandas investigated the cycles of maternal activity levels in the wild [60] and in captivity [64]. These studies provided insight into the circadian and circannual cycles of maternal female activity, and allowed for comparison to other adults and females, but were not able to observe maternal behaviors in detail, and for the captive study, were not able to capture nighttime activity. Our study is able to expand on the knowledge of maternal behavior in captive pandas to assist zoos in understanding mother and cub dynamics to best provide for their care.

Our results do not show a significant difference in the maternal female’s activity compared to other females, which is a similar result to what was found by Mainka and Zhang [64] in maternal females whose daytime activity cycles across four seasons did not show a significant difference either. However, our results did show the maternal female displaying consistently higher activity levels than the other females in both her circadian and circannual rhythm. Though not significant, this increased activity could be a signal of maternal care, since the mother must sustain herself as well as display other maternal behaviors that the non-maternal females do not display.

The circadian rhythms of new mammalian mothers are often disrupted in response to having an infant who is still developing a circadian rhythm and nursing. Although most mammal newborns, aside from cetaceans, have increased sleep and reduced mobility after birth that are associated with requirements for brain and body development [2], mothers must still be increasingly attentive to their newborn, displaying new behaviors that are only associated the with care of offspring. In our study, we found that the cub did indeed show reduced activity compared to the pandas in other life stages, showing more rest and less feeding, which is a result that was also found in cubs in the Wolong Reserve [64].

The full circadian cycle of mother and cub behaviors, including maternal and nursing behaviors, has not been previously investigated in captive pandas. This was an opportunity to investigate how the circadian rhythm of a cub develops in captivity, how a mother’s circadian rhythm changes or not as a result of having a nursing cub, and how the levels of nursing and other maternal behaviors shift over the first years of a cub’s life. In mammals, circadian rhythms begin to develop slowly before birth, largely emerging in different cell types at stages of cellular differentiation [86]. After birth, there has been evidence in rats and humans that the circadian rhythm is partially entrained by the light/dark cycle and partially by the mother’s cycle [87,88,89]. In rats, it was also found that certain clock genes shift after weaning, including in peripheral organs like the liver [87,90]. These studies suggest that a nursing mother can likely have effects on the development of an infant’s circadian rhythm and that nursing behavior itself plays an important role in regulating shifts. Though we did not measure causality, we did see evidence of the mother and cub displaying similar cycle lengths of behavior in their wavelet transforms (Figure 9c,d). The main difference was seen in the mother presenting a 24 h cycle with increased magnitude towards the end of the observations, while the cub had a 24 h cycle with low magnitudes throughout the study. This is consistent with our predictions that the cub would not have a consistent circadian rhythm in the first stage of life and that the mother would potentially have a deregulated circadian rhythm while nursing and would slowly regain a consistent circadian cycle. This may be further supported by the synchronization in their 24 h cycles when the mother did not have a strong 24 h cycle, potentially indicating that the mother was synchronizing to an irregular 24 h cycle that the cub possessed.

Our results showing that the mother and cub did not have changes in the levels of nursing throughout the course of observation is expected, as the cub was still dependent by the time the observations ended. The keepers began creating independence by gradually housing the mother and cub separately beginning 2 months after the observations ended. Occasional nursing was seen by the keepers until the separation process was completed, and the mother and cub were housed entirely separate 11 months after the study concluded. However, another study on a giant panda’s maternal behavior found a gradual decrease in non-nursing maternal behavior towards her cub up to 150 days after birth [91]. We also saw this pattern over time, though firm conclusions cannot be drawn because of the inability to model these behaviors. We expected that the proximity between the mother and cub would match this result and decline overtime, as was also seen in the study by Zhang [91]. However, the proximity data across the seasons do not show an increasing proportion of time spent in distant proximity between the mother and cub, but rather show an increase in proximity towards the end of the observation period. Without data on “approach” and “leave”, we are unable to determine who is regulating the proximity. Interestingly, nursing and other maternal behaviors were mainly observed in the night. The circadian cycle of proximity matched this, where the mother and cub were proximate and in contact mainly after working hours. Further investigation should be conducted to determine if this is a natural cycle of nursing behavior in pandas, or whether this was only observed because nighttime allowed for undisturbed interactions between the mother and cub in the absence of zoo visitors and staff. Giant pandas are notoriously difficult to breed, and even following a live birth, cubs may be rejected both in the wild and in captivity [91]. Therefore, understanding the rhythms of maternal behaviors can assist staff in providing the proper environment and privacy at the right times of day and across the time spent nursing, helping giant panda mothers successfully raise their offspring.

## 5. Conclusions

Our study demonstrates a holistic approach in assessing animal behavior, their needs, and welfare indicators through the use of circadian and circannual rhythms. Circadian and circannual rhythms provide a fuller view of what a healthy, adaptive cycle of behavior is, and since circadian clocks regulate internal physiological clocks, the rhythms can also provide insight into the physiological state of the animal. Using this approach to study captive giant panda behavior allowed us to understand the possible energetic dynamics between behaviors like locomotion, drinking, feeding, and resting. It is pleasing to report that the natural cycle of the three activity peaks in the captive giant pandas are similar to their wild counterparts. These results also demonstrate how this method can be used to determine species’ typical levels of behaviors to validate valences applied to behaviors in relation to welfare. In addition, with this broad view, we were able to highlight some possible links between the timing of migration in the wild and stereotypical/abnormal behavior, sexual-related behaviors, and feeding behaviors in captivity, suggesting that this is how thwarted migration may manifest. We also expanded on the knowledge of how a mother and cub interact in captivity, adding important context to the kinds of maternal behaviors displayed and the times of day that are more likely to show the mother and cub interacting. Exploring sex and life stage in concordance with rhythms adds another layer of context. Investigating all these variables together, including life stage, sex, time of day, and time of year, is more informative than investigating any of them individually, because connecting them allows for new associations and predictable changes in rhythm by sex or life stage to be identified.

Circadian clocks are evolutionarily adaptive and are at the basis of biological function in living organisms. This makes circadian rhythms equally relevant in wild individuals as in captive ones, since the rhythms persist in captive environments and regulate many processes affecting welfare. We must be conscious of circadian rhythms as a way to educate ourselves on the evolutionary needs of an animal. This method can be modified in accordance with each zoos’ questions to investigate the welfare states of animals. The insight on animal behavior that is gained from these kinds of analyses can be used by zoos to modify their practices of care in accordance with the biological needs of a species, and can also allow them to predict what resources or special care might be needed across the day, year, and lifespan of the animals.

## Figures and Tables

**Figure 1 animals-13-02401-f001:**
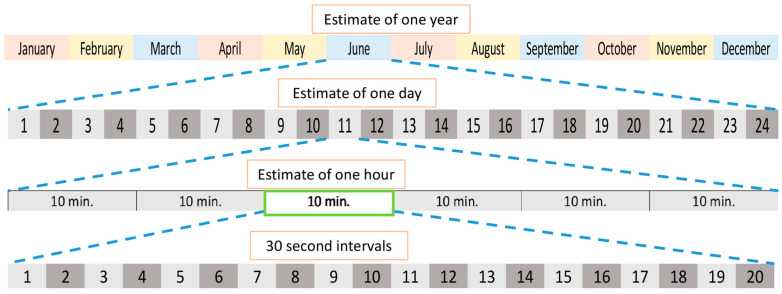
Representation of the sampling method used to obtain estimates of circadian cycles and a circannual cycle. Each month, observations began on the 10th of the month and continued through the end of the month until one 10 min session was completed for each hour of the day. For pandas observed on cameras without night vision, observations continued until a 10 min session was completed for each daylight hour. Observer availability and panda visibility determined when the sessions were completed.

**Figure 2 animals-13-02401-f002:**
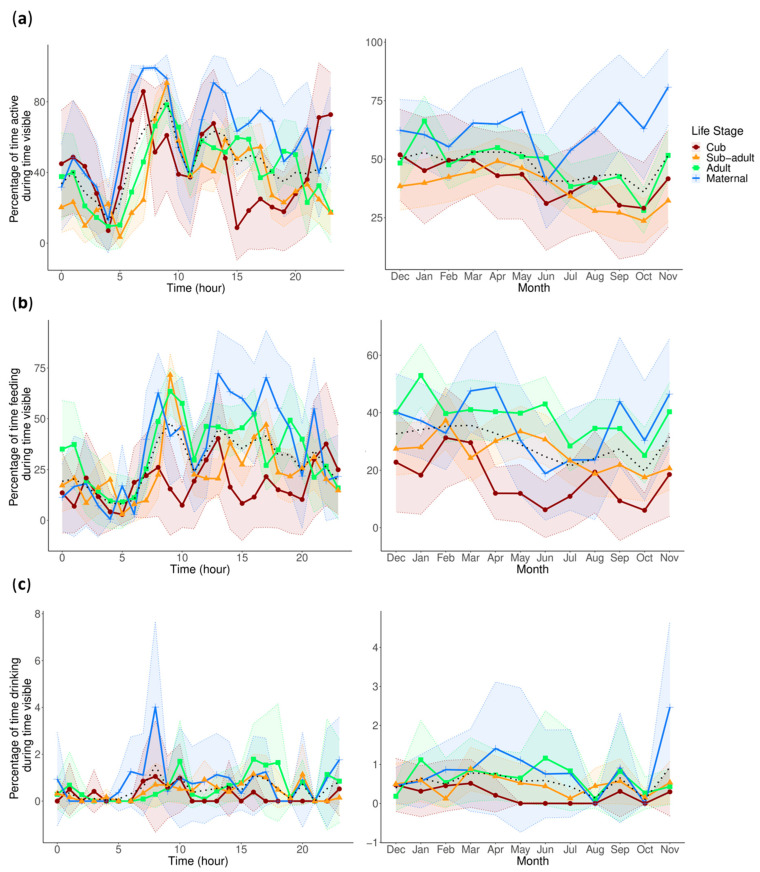
Circadian and circannual cycles of activity (**a**), feeding (**b**), and drinking (**c**) by life stage (cub, *n* = 1 for 24 h; subadult, *n* = 5, 3 for 24 h, 2 for daylight; adult, *n* = 6, 2 for 24 h, 4 for daylight; maternal, *n* = 1 for 24 h). Behaviors are displayed as estimated percentages of time active/displaying behavior while in sight, controlling for time out-of-sight, and are averaged by hour (circadian) or month (circannual). The black dotted line represents the mean across all individuals. Scales are not standardized between circadian and circannual graphs. The shaded regions are the 95% confidence intervals for the activity.

**Figure 3 animals-13-02401-f003:**
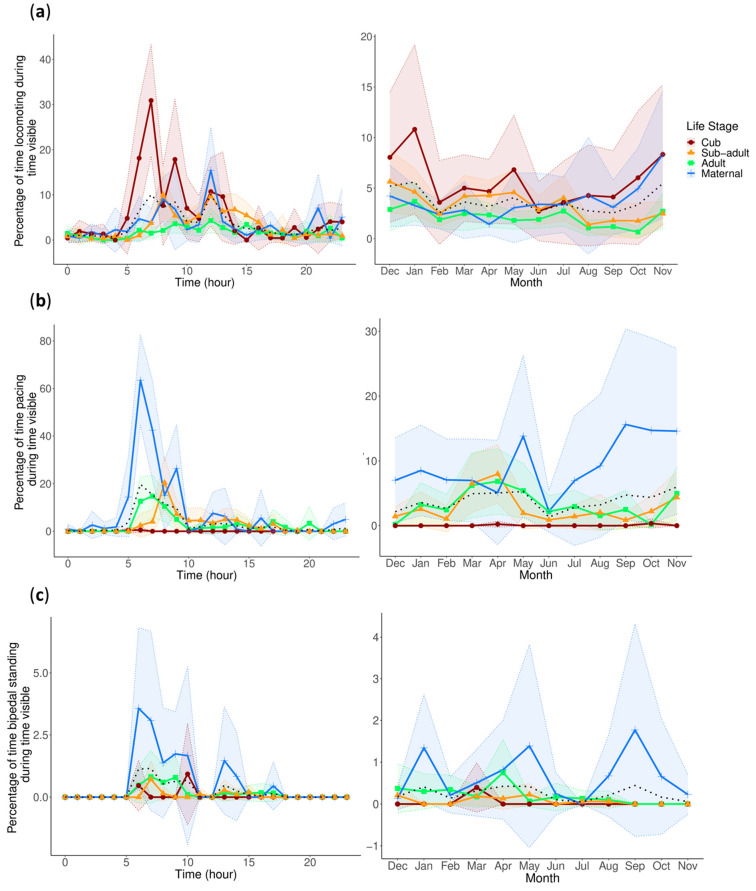
Circadian and circannual cycles of locomotion (**a**), pacing (**b**), and bipedal standing (**c**) across life stages (cub, *n* = 1 for 24 h; subadult, *n* = 5, 3 for 24 h, 2 for daylight hours; adult, *n* = 6, 2 for 24 h, 4 for daylight; maternal, *n* = 1 for 24 h). Behaviors are displayed as estimated percentages of time displaying behavior while in sight, controlling for time out-of-sight, and are averaged by hour (circadian) or month (circannual). The black dotted line represents the mean across all individuals. Scales are not standardized between circadian and circannual graphs. The shaded regions are the 95% confidence intervals for the activity.

**Figure 4 animals-13-02401-f004:**
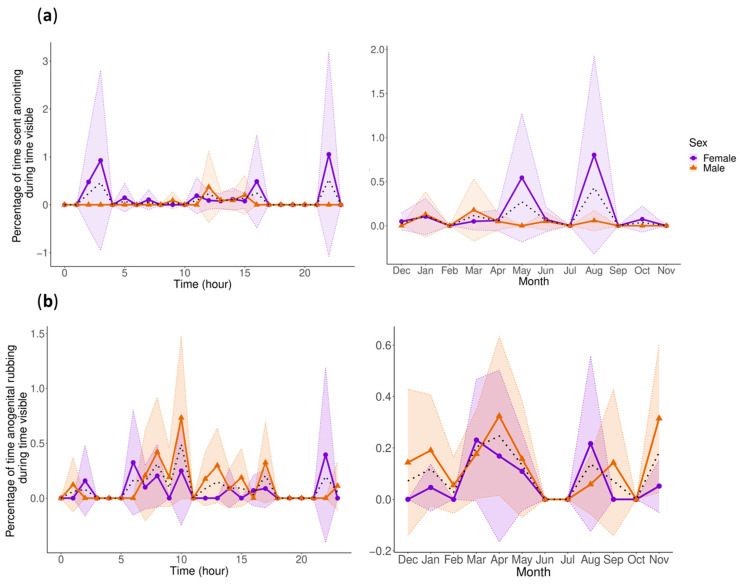
Circadian and circannual cycles of scent anointing (**a**) and anogenital rubbing (**b**) by sex (male, *n* = 5, 4 for 24 h, 1 for daylight; female, *n* = 6, 3 for 24 h, 3 for daylight). Behaviors are displayed as estimated percentages of time active/displaying behavior while in sight, controlling for time out-of-sight, and are averaged by hour (circadian) or month (circannual). The black dotted line represents the mean across all individuals. Scales are not standardized between circadian and circannual graphs. The shaded regions are the 95% confidence intervals for the activity.

**Figure 5 animals-13-02401-f005:**
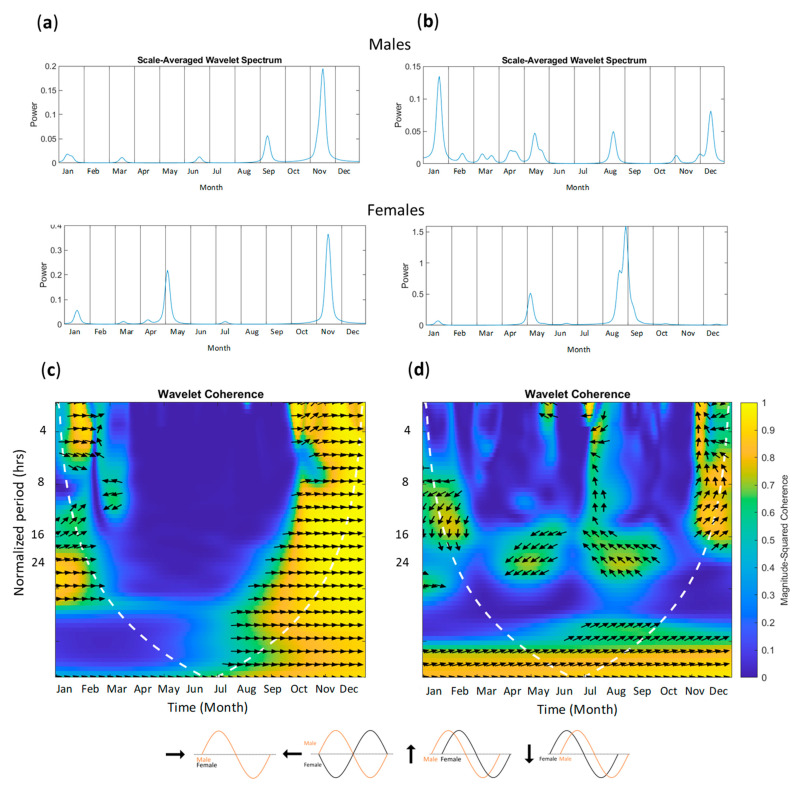
Wavelet coherence analyses of sexual-related behaviors between breeding pairs at Zoo D (**c**) and Zoo B (**d**). Zoo D has a sub-adult female and an adult male, and Zoo D has a male and female sub-adult pair. The dashed white line indicates a 5% significance level. The *x*-axis is the time of year in months, the *y*-axis is the normalized frequency between the two signals, and the color represents the strength of the correlation (yellow is high, dark blue is low, scale is on right hand side). The kind of phase relationship between the signals are noted by the arrows (refer to arrow key). We also display the scale-averaged wavelet spectrums for each individual panda at Zoo D (**a**) and Zoo B (**b**). These can be seen as smoothed circannual rhythms of the circadian rhythms of all sexual-related behavior. The *x*-axis is the time in months, which matches with the coherence figures, and the *y*-axis is the power (average magnitude across all period lengths for that time).

**Figure 6 animals-13-02401-f006:**
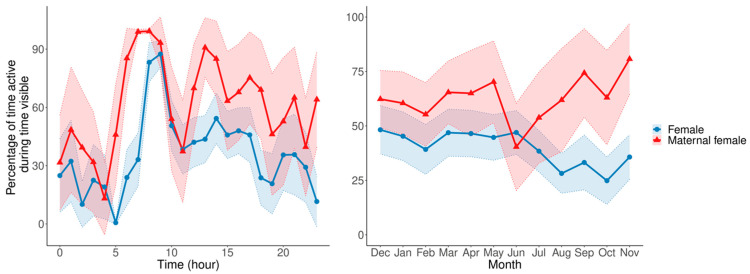
Circadian and circannual cycles of activity between the females (*n* = 5, 2 for 24 h, 3 for daylight) and the mother (*n* = 1 for 24 h). Behaviors are displayed as estimated percentages of time active while in sight, controlling for time out-of-sight, and are averaged by hour (circadian) or month (circannual). Scales are not standardized between circadian and circannual graphs. The shaded regions are the 95% confidence intervals for the activity.

**Figure 7 animals-13-02401-f007:**
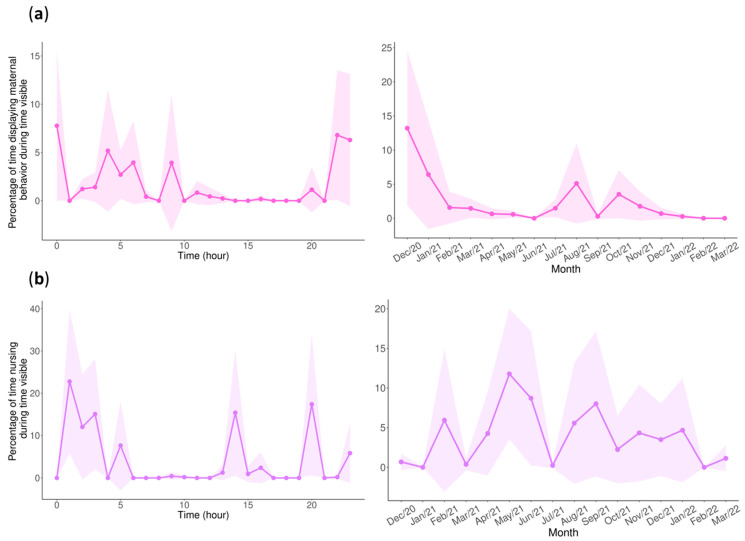
Circadian and circannual cycles of non-nursing maternal behavior (**a**) and nursing (**b**) for mother and cub. Behaviors are displayed as estimated percentages of time displaying behavior while in sight, controlling for time out-of-sight, and are averaged by hour (circadian) or month (circannual). Scales are not standardized between circadian and circannual graphs. The shaded regions are the 95% confidence intervals for the activity. Data are averaged between mother and cub.

**Figure 8 animals-13-02401-f008:**
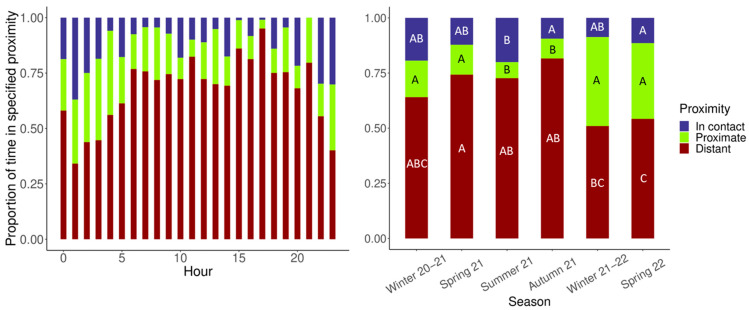
Proximity (in contact, blue; proximate, green; distant, red) of mother and cub displayed as estimated proportions of time by hour and by season. The letters within the proximity categories represent significance. Within each proximity category, different letters represent a significant difference between seasons for that proximity. Seasons that share any letter for a proximity category are not significantly different. Time when mother and cub were separated by zookeepers was excluded so that only proximity when mother and cub had a choice is displayed. Therefore, proportions of time are extended to 1 excluding keeper separation, which are only displayed during working hours of keepers.

**Figure 9 animals-13-02401-f009:**
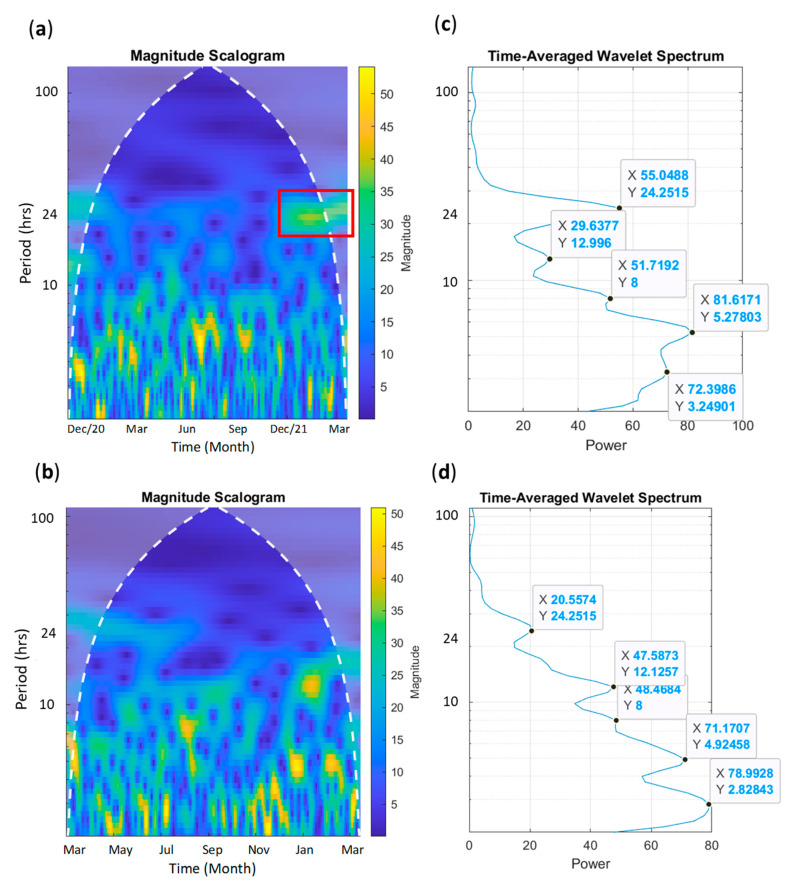
Continuous wavelet transform analysis of mother (**a**) and cub (**b**) overall activity. The *x*-axis represents the time in months across the observation period. Therefore, the mother has three more months of observations than the cub at the start. The *y*-axis is the period length in hours; periods above 24 h do not have real-world equivalents given our sampling method. The magnitude is represented with color. The red square in the mother’s wavelet highlights when her 24 h period of activity was more consistent. Time-averaged wavelet spectrums are also presented for mother (**c**) and cub (**d**). These graphs represent the average magnitude of period lengths across the observation period. The *x*-axis is the power (magnitude over time), and the *y*-axis is the period length, which equates to the period lengths in the wavelet.

**Figure 10 animals-13-02401-f010:**
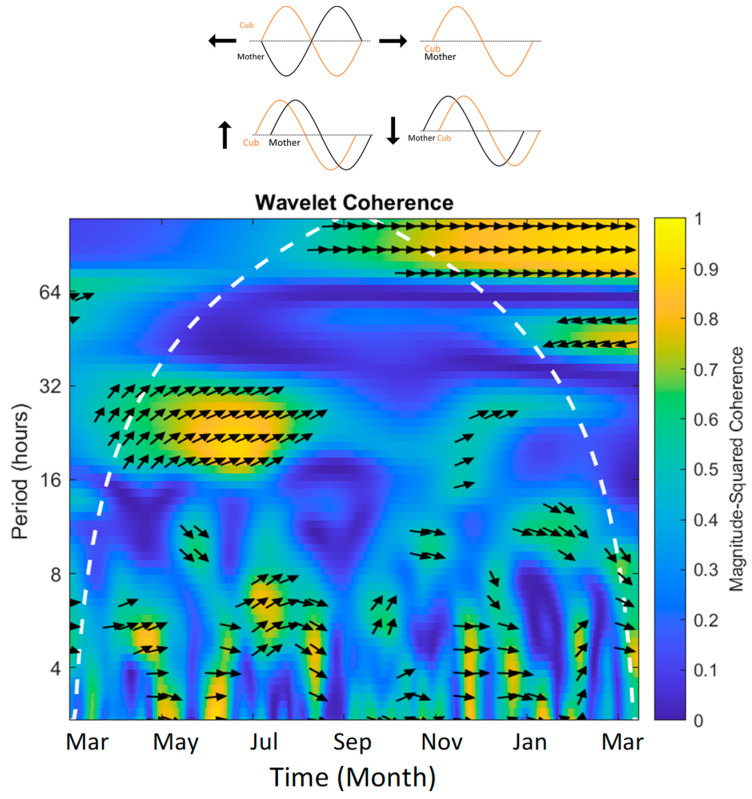
Wavelet coherence analysis of activity between the mother and cub pair. The *x-*axis is the time of year in months, the *y*-axis is the normalized frequency between the two signals, and the color represents the strength of the correlation (yellow is high, dark blue is low, scale is on right hand side). The kind of phase relationship between the signals is noted by the arrows (refer to arrow key). Data presented in this wavelet exclude the data collected for the mother in Winter 20–21.

**Table 1 animals-13-02401-t001:** Information on the study animals and camera visibility. Within Zoo C, sub-adults could be distinguished from adults, but individuals could not be identified. Therefore, these individuals are listed as 5a, 5b, 6a, and 6b. Individuals with the same number had their data combined, and for individuals 6a and 6b, the sex was categorized as unknown in analysis. Mating opportunity refers to when pandas are normally given opportunities by keepers to breed naturally or through artificial methods.

Zoo	Panda	Sex	Life Stage	Feeding Schedule	Mating Opportunity	Camera Visibility
A	1	F	Adult	On average, 9 (female) or 10 (male) times a day between 0700 and 1800 h	Around March; either natural or via artificial insemination	Daylight
	2	M	Adult			
B	3	F	Sub-adult	Unknown	Breeding pair, but unknown time and method	24 h
	4	M	Sub-adult			
C	5a	F	Sub-adult	First bamboo feed by 0900 h, second at 1200 h, third between 1330 and 1430 h, and final between 1530 and 1630 h	None	Daylight
	5b	F	Sub-adult			
	6a	F	Adult		Post-reproductive	
	6b	M	Adult			
D	7	F	Sub-adult	First bamboo feed between 0745 and 0845 h, second before 1200 h, and final between 1300 and 1700 h	Around March; natural	24 h
	8	M	Adult			
E	9	M	Adult	First bamboo feed between 0800 and 1000 h, second around 1300 h, and final between 1700 and 1800 h	Castrated for medical reasons	24 h
F	10	F	Maternal	Bamboo provided approximately 5× per day, with first feeding at 0730 h and final feeding between 1330 and 1400 h	N/A	24 h
	11	M	Cub			

**Table 2 animals-13-02401-t002:** Summary of ZINB models for activity, sexual-related behavior, and stereotypical/abnormal behavior. N/A appears for models that did not run (N/A across row), likely due to overparameterization, or models for which the best fit was likely not found (N/A for AIC and BIC).

Behavior Category	Regression Type	Iterations	AIC	BIC	Df (Residual)	Dispersion Parameter
Activity	Linear	58	13,484.6	13,631	2566	3.22
Resting/sleeping	Quadratic	60	15,680	15,826.4	2566	3.7
Feeding	Linear	57	10,541.2	10,687.6	2566	2.56
Drinking	Linear	51	1456	1602.4	2566	0.238
Locomotion	Quadratic	49	4594.3	4740.7	2566	0.33
Pacing	Quadratic	54	2421.1	2567.5	2566	2
Bipedal standing	Quadratic	49	580.9	727.3	2566	0.33
Scent anointing	Quadratic	55	298	444.4	2566	2.52
Anogenital rubbing	Quadratic	40	434.9	546.2	2569	0.224
Female activity	Quadratic	52	6407.2	6493.8	1192	4.82
Maternal	N/A	N/A	N/A	N/A	N/A	N/A
Nursing	Quadratic	38	613.1	688.7	621	1.18
In contact	Quadratic	40	1722.6	1798.2	621	1.07
Proximate	Quadratic	41	2310.6	2386.2	621	1.67
Distant	Linear	38	N/A	N/A	621	1.12

**Table 3 animals-13-02401-t003:** Summary of significant and potentially significant (*p* < 0.1) coefficients from the ZINB models on general behaviors. In the conditional model, (+) coefficients indicate that the category level/variable is larger than the grand mean, and a (−) coefficient means it is smaller than the grand mean. For zero-inflated coefficients, (+) indicates that the category level/variable is more likely to be a zero, and (−) means it is less likely to be zero (i.e., more likely to be a positive integer). Variables with an asterisk did not reach the significance threshold but did show possible trends towards significance, with *p* < 0.1.

	Conditional Model				Zero-Inflated Model			
Behavior	Variable	Coefficient	Z-Value	Pr (>|z|)	Variable	Coefficient	Z-Value	Pr (>|z|)
Activity	Summer	−0.057	−2.250	0.025	Summer	0.155	2.213	0.027
	Winter	0.063	−2.250	0.025	Autumn	0.213	2.850	0.004
	Cub	−0.105	−2	0.046	Spring	−0.320	−4.540	<0.001
	Maternal	0.118	2.390	0.017	Maternal	−0.481	−1.984	0.047
					Hour	−0.016	−2.410	0.016
Resting/sleeping	Spring	−0.064	−2.674	0.008	Spring	0.214	2.926	0.003
					Autumn	−0.314	−3.737	<0.001
					Summer	−0.209	−2.687	0.007
					Winter	0.310	4.213	<0.001
					Hour	0.020	2.815	0.005
					Adult	0.197	2.231	0.026
					Cub	−0.506	−3.797	<0.001
					Maternal	0.697	5.925	<0.001
					Sub-adult	−0.387	−4.628	<0.001
Feeding	Adult	0.177	2.590	0.010	Autumn	0.221	2.886	0.004
	Cub	−0.376	−3.370	<0.001	Winter	−0.189	−2.683	0.007
	Hour	0.007	2.460	0.014	Spring	−0.244	−3.534	<0.001
	Winter *	0.045	1.800	0.072	Summer	0.212	2.950	0.003
					Hour	−0.030	−4.565	<0.001
					Cub *	0.403	1.801	0.072
Drinking	Cub *	−2.094	−1.813	0.070	Spring	−0.536	−2.268	0.023
					Summer	0.532	2.345	0.019
					Hour *	−0.043	−1.761	0.078
					Male *	0.459	1.727	0.084
Locomotion	Winter	0.223	2.545	0.011	Spring	−1.040	−2.710	0.007
	Summer	−0.199	−2.163	0.031	Sub-adult	0.953	2.205	0.027
	Hour	−0.071	−5.932	<0.001	Hour	−0.494	−7.104	<0.001
	Adult *	−0.559	−1.774	0.076	Winter *	0.570	1.840	0.066
					Cub *	−0.869	−1.805	0.071
Pacing	Spring	0.251	2.460	0.014	Spring	−0.365	−2.937	0.003
	Maternal	1.546	1.999	0.046	Summer	0.298	1.970	0.049
	Hour	−0.041	−2.058	0.040	Hour	0.051	3.890	<0.001
	Cub *	−3.565	−1.750	0.080				
Bipedal standing	Spring	1.071	3.265	0.001	Winter	−1.373	−2.586	0.010
	Hour	0.400	5.028	<0.001	Autumn	1.383	1.916	0.055
					Hour	0.631	6.423	<0.001
Scent anointing	Hour	0.397	4.822	<0.001	Hour	0.790	3.098	0.002
	Male	−1.477	−2.028	0.043	Summer *	−1.940	−1.710	0.087
Anogenital rubbing	Spring	1.054	2.510	0.012	Autumn	2.489	2.218	0.027
	Summer	−3.023	−4.047	<0.001	Summer	−5.262	−2.467	0.014
	Hour	0.221	3.406	0.001	Hour	0.412	3.406	<0.001
	Female *	0.973	1.823	0.068	Female	2.056	2.406	0.016
					Male	−2.680	−2.799	0.005
					Winter *	1.900	1.850	0.064

**Table 4 animals-13-02401-t004:** Summary of significant and potentially significant (*p* < 0.1) coefficients from the ZINB models of female activity, maternal behaviors, and mother and cub proximity. In the conditional model, (+) coefficients indicate that the category level/variable is larger than the grand mean, and a (−) coefficient means it is smaller than the grand mean. For zero-inflated coefficients, (+) indicates that the category level/variable is more likely to be a zero, and (–) means it is less likely to be zero (i.e., more likely to be a positive integer). Variables with an asterisk did not reach the significance threshold but did show possible trends towards significance, with *p* < 0.1. “None” signifies that no coefficients reached significance thresholds in that model.

	Conditional Model				Zero-Inflated Model			
Behavior	Variable	Coefficient	Z-Value	Pr (>|z|)	Variable	Coefficient	Z-Value	Pr (>|z|)
Female activity	Maternal female *	0.105	1.750	0.08	Maternal female *	−0.684	−1.672	0.0945
Maternal behaviors	N/A				N/A			
Nursing	None				Hour	0.059	2.370	0.018
In contact	Spring 22	−0.481	−2.030	0.0420	Spring 22	−0.630	−1.995	0.046
	Summer 21	0.534	3.185	0.001				
Proximate	Autumn 21	−0.618	−4.473	<0.001	Spring 21	0.557	2.918	0.004
	Spring 22	0.448	2.968	0.003	Spring 22	−0.687	−2.398	0.016
	Summer 21	−0.594	−4.163	<0.001	Summer 21	0.415	2.065	0.039
					Hour	0.059	4.333	<0.001
					Winter 20–21 *	0.436	1.802	0.072
Distant	Autumn 21	0.067	1.980	0.047	Autumn 21	−1.161	−3.802	<0.001
	Spring 21	0.091	2.740	0.006	Winter 20–21	0.640	2.802	0.005
	Spring 22	−0.161	−2.700	0.007	Hour	−0.044	−3.026	0.002
	Summer 21 *	0.061	1.790	0.073				

## Data Availability

The data are available through DataSTORRE through the link http://hdl.handle.net/11667/208 (accessed on 5 May 2023) and a second link that will include the mother and cub data. The mother and cub data are still being reviewed by DataSTORRE, but the link will be provided once the data are accepted.

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
