# Peer review of "Understanding Circadian and Circannual Behavioral Cycles of Captive Giant Pandas (Ailuropoda melanoleuca) Can Help to Promote Good Welfare"

_animals, 2023, doi:10.3390/ani13152401_

Round 1

Reviewer 1 Report

This is a very interesting study and wonderful paper. The authors used the video clips of giant pandas from six zoos/institutions. They tried to reveal the circadian and circannual behavioral cycles of 13 captive giant pandas including a female and a cub. They creatively used the methods of the zero-inflation negative binomial modelling and post hoc pairwise comparisons and continuous wavelet transform and wavelet coherence analysis analyzing the data of each behavioral category. The results are novel and interesting, the underlying meaning of their results are greatly helpful for animal management and welfare. This manuscript has been well written. However, I have some concerns about this manuscript before I recommend an acceptance.

First, also the biggest one, is the sample size of their study. They had only 6 male (including one cub) and 7 female subjects with different ages in their study. Among them, the sex of the adults at Zoo C was not identified which further caused the valid sample size for analyses to be reduced again. There is only one pair of mother and cub. Moreover, camera recordings at night were not available for pandas at Zoo A and Zoo C. Thus, it is inappropriate to extrapolate their results too much even though I know those data are merit for authors.

Second, those pandas are located in six institutions. I am not sure the detailed information about those zoos. Thus, I don’t know whether the authors have realized the possible effects of the latitude of those pandas/zoos location on the circadian of those pandas. Moreover, bamboos and other associate foods usually are provided by zoo staff, and the food delivery schedule might vary with different zoos even though there is a standard protocol for those loaned pandas. Thus, the rhythm of feeding of those pandas might be differ with different zoos. It might be better to explain this in the discussion section.

Third, it is true that the age of the panda showing different circadian which has been previously reported by Mainka and Zhang (1994). However, I think it might be precise and informative if you can use the real age of the panda rather than the age group in your analysis.

Fourth, it might be better to provide detailed information about those pandas name or studbook numbers and their ages, and the zoo names in Table 1. It is necessary and important for reader’s better reference and evaluation.

Fifth, I would recommend the author reduce the introduction and discussion section, especially the discussion section. I think it might be better to stick to your own main results and findings, and delete the irrelevant or untested results.

Last, I would suggest the author remove the contents of mother and cub by closely concentrating your study on the circadian of captive pandas even though I know that will be a hard choice for you. It seems to me that this section has diluted your main findings in this manuscript.

 In summary, this is a very interesting paper. The quality will be greatly improved if the authors can closely concentrate on the main topic and main findings.

Thus, I recommend a major revision and I am looking forward to its coming out.

Reviewer 2 Report

Questions and comments:

1)      In the first paragraph of the introduction, the authors propose an approach to evaluate the four domains of animal welfare (nutrition, environment, health, and behavior) in connection with the diel and annual basis for the subsequent interpretation of the 5th domain - the affective state. The affective state is directly related to the psychological mechanisms controlled by the individuals. However, it is important to note that the arguments presented by the authors often focus on the general fitness of the animal at the species level rather than considering individual mental needs and conditions. For instance, when incorporating nutrition and behavior in pandas, the authors argue in terms of the overall fitness of the animal based on optimal foraging theory. This theory does not explicitly address the psychological mechanisms controlled by individual animals but instead discusses how natural selection shapes behavior to maximize fitness within the environment and energy constraints. It is necessary to highlight that while natural selection optimizes the fitness of organisms, it does not guarantee their well-being. Fitness and animal welfare are related concepts, but they are not synonymous. Animal welfare encompasses not only an animal's physical health but also its mental and emotional well-being, which may not always align with what maximizes fitness from an evolutionary perspective.

2)      Specify the principle used for categorizing various forms of behavior into three valencies of an affective state in Table A1 and how it is discussed in the paper.

3)      Have you compared the feeding activity of the cubs by summing up the activity of feeding + nursing?

4)      How often are pandas fed during the day in these zoos?

5)      Was a separate model built for sleep as a target variable? In the discussion, you only write about describing the activity, as opposed to sleep, which is referred to in the figure. However, I'm missing the sleep analysis in the Results section and information on the factors that affected sleep in your study.

6)      The volume of collected material is not entirely clear. Was your goal to collect only one 10-minute observation session for each hour of the day, or should each hour be fully covered by 10-minute sessions? Were there any breaks between sessions? Have there been sessions where no breaks occurred between them? It would be helpful to include information about the degree of randomness in breaks between observation sessions to demonstrate the level of independence between them.

 Additionally, the phrase is not entirely clear: “the 10-minute sessions were completed depending on the availability of an observer ”. Does this imply that unfinished sessions were included in the analysis?

6 ) Please clarify the role of the observers. Do they assist in data collection for multiple or per one panda?

Introduction

Line 46-47. Please provide a reference for the statement and give an example of what wild animals anticipate within a day and within a year.

Line 49-50: There are studies, for instance, on elephants and gorillas, investigating animal behavior during nighttime and in different seasons. It is preferable to rephrase this sentence to indicate that the research is focused on a limited number of species.

Line 52: Kindly provide specific references to support this statement other than the one mentioned in this review. Please be more precise than simply referring to a review.

Line 57: Replace "the affective states of species" with "the needs of animals within a species." The article does not discuss interpretations of animals' affective states. Using "the affective states of species" implies that all individuals within a species share a single affective state, which may not be the case due to potential variations in affective states among individuals.

Lines 58-59: Please provide a clearer connection between the 1st and 2nd paragraphs. Currently, it is unclear how understanding animal cycles, their energy needs, and energy maximization will help in interpreting the affective states of species.

Line 59: Please define the term "energy maximization" and provide a specific reference.

Lines 76-77: Please provide a specific reference for this statement.

Lines 100-125: Add a sentence explaining how breeding is related to animal welfare, not just breeding success.

Lines 126-129: Please include other important sleep features such as restorative sleep, as sleep deprivation can cause severe visceral dysfunctions and ultimately lead to the unavoidable death of animals: (Pigarev, I. N., & Pigareva, M. L. (2015). The state of sleep and the current brain paradigm. Frontiers in Systems Neuroscience9, 139.).

Line 151-152. Replacefat storages” with “fat storages accumulated before the migration season”.

Line 161. Replace “provide insight into the welfare state of the animal” with “provide insight into the animal needs”.

Line  171-172. Please, replace the word "ideal" with a synonym for duplicates.

Material and Methods

Line 209. Replace “panda mother” with “panda dam”.

Table 1. Describe the “Mating opportunity” column in the table heading.

Table A1. Please, describe how the categories of affective states were selected.

Please rephrase “same as stereotypic pacing, except animal need not take the same path 3 or more times in a row” to “same as stereotypic pacing, except animal travel in an irregular pattern 3 or more times in a row”.

Line 225-236. Please, provide information about the volume of collected data.

Line 262. Please remove the gray highlighting for this reference.

Line 292-293. Replace “would account for zero-inflation and the resulting overdispersion ” with “would account for zero-inflation observed due to a combination of factors, and the resulting overdispersion in the data”.

Lines 314-315: Please provide a summary of how latitude, temperature, and the amount of daylight influenced behavioral cycles to facilitate a discussion of the results of this study. 

Lines 357-360: Please provide clarification on the specific aspects you analyzed to address the question regarding the synchronization of mother and cub cycles.

Line 369: Please add a link to Table A1. Additionally, note that in the ethogram, this category is referred to as "sexual behavior" and not "sexual-related behaviors.

Results

"Graph – Is it possible to increase the DPI to avoid grain?

Table 3: Standardize the table style for consistent decimal precision.

Line 412: Replace "activity showed a decrease" with "activity showed a slight decrease."

Line 413: Replace "an increase in the Winter" with "a slight increase in the Winter when compared to the grand mean of all seasons."

Line 497. Replace “compared to the grand mean” with “compared to the grand mean observed across both males and females”.

Line 524-525. Add the reference to Table A1 after the phrase “sexual-related behaviors”.

Figure 5: Replace the phrase 'Wavelet coherence analyses between' with 'Wavelet coherence analysis of sexual-related behaviors between'. The figure displays captions a-d in lowercase letters and the figure caption in uppercase letters. Please maintain consistency. The Y-axis in the figure labeled 'females (a)' is partially obscured.

"Table 4: Standardize the table style for consistent decimal precision.

Replace "Summary of significant (or near significant) coefficients" with "Summary of significant and potentially significant (p<0.1) coefficients."

Replace "Winter20-21*" with "Winter 20-21*."

Describe the "None" result in the table caption/heading.

Figure 6: Note that in Line 210, the category 'mother' was assigned to 'a maternal life stage,' whereas here it is assigned to a different sex.

Figure 8: Provide a clearer description of the significant letters.

The letters "C" and "D" are not shown in the figure. In the figure caption, the letters "C" and "D" are capitalized, while "a" and "b" are lowercase.

Discussion

Line 662-664: Please rephrase the sentence to clarify the contrast between this statement and the previous one. It is unclear whether you are contrasting information about three peaks of activity or about crepuscular behavior, which was not previously discussed. Additionally, provide a reference for the statement that states, "are not crepuscular, as was assumed before."

Please provide the specific number of animals for the phrase “completed on wild pandas”.

Line 752. Replace “had reduced sexual competitive drives” with “had reduced sexual competitiveness”.

Line 756-757.  Add support reference for the “competitive signaling hypothesis”.

Line 813. Rephrase “instincts” with “motivations”.

Line 825. Add “, respectfully.” after “in Summer”.

Line 845-847. Here and after, replace “feeding anticipatory activity (FAA)” with “food anticipatory behavior”. “Food anticipatory behavior" is a more commonly used and preferred term. It highlights the active and goal-directed nature of the behavior, emphasizing that animals are anticipating reward rather than merely exhibiting activity.

Please rephrase the sentence "One possible explanation for the distinct rhythmic pattern observed in an early morning peak of both pacing and bipedal standing (which are anticipatory behaviors displayed near keeper doors) could be food anticipatory activity (FAA)." It is difficult to understand what "(an anticipatory behavior displayed at keeper doors)" refers to in this sentence.

Line 851-852. Provide a reference for the statement: “Since stereotypic/abnormal behaviors may be a sign of anticipation”.

Line 950. Replace “welfare” with “their needs”.

Reviewer 3 Report

In this study, the authors have reported the circadian and circannual behavioral patterns in captive pandas. The study methods are very detailed and clear, data collections are satisfactory and statistical approaches are well-planned for concluding their findings. However, the only drawback is the study entirely focused on captive pandas. They took six zoos of which one zoo (E) has only a single castrated adult male panda. Since the natural free-ranging population experiences varied types of intra and inter-specific interferences, captive pandas, especially who stay all alone (for eg. Zoo E) are expected to behave differently due to anthropogenic stress. Therefore, it would be great if the authors target at least a few free-ranging natural populations (non-captive) for their long-term study which could provide important insight into panda’s actual circadian behavioral rhythms.
